# In Situ Electrochemical Formation of Oxo-Functionalized Graphene on Glassy Carbon Electrode with Chemical Fouling Recovery and Antibiofouling Properties for Electrochemical Sensing of Reduced Glutathione

**DOI:** 10.3390/antiox12010008

**Published:** 2022-12-21

**Authors:** Chunying Xu, Gang Li, Liju Gan, Baiqing Yuan

**Affiliations:** School of Chemistry and Materials Science, Ludong University, Yantai 264025, China

**Keywords:** antioxidant, reduced glutathione, electrochemical sensor, oxo-functionalized graphene, electrochemical oxidation and reduction, antifouling

## Abstract

Electrochemical detection can be used to achieve intracellular or in vivo analysis of reduced glutathione (GSH) in tissues such as brain by using a microelectrode, which can help to better understand the complex biochemical processes of this molecule in the human body. The main challenges associated with electrochemical GSH detection are the chemical fouling of electrodes, caused by the oxidation product of GSSG, and biofouling due to the non-specific absorption of biological macromolecules. Oxo-functionalized graphene was generated in situ on the surface of a glassy carbon electrode using a green electrochemical method without using any other modifiers or materials in a mild water solution. The fabricated oxo-functionalized graphene interface was characterized by Raman spectroscopy, X-ray photoelectron spectroscopy, electrochemistry, electrochemical impedance spectroscopy, and contact angle measurements. The interface showed high electrocatalytic activity towards the oxidation of GSH, and a simple and efficient GSH sensor was developed. Interestingly, the electrode is reusable and could be recovered from the chemical fouling via electrochemical oxidation and reduction treatment. The electrode also exhibited good antibiofouling properties. The presented method could be a promising method used to treat carbon materials, especially carbon-based microelectrodes for electrochemical monitoring of intracellular glutathione or in vivo analysis.

## 1. Introduction

Reduced glutathione (GSH, l-glutamyl-l-cysteinyl-glycine), a tripeptide composed of amino acids glutamate, cysteine, and glycine, is the most abundant low-molecular-weight thiol in mammalian cells with molar concentrations [1]. GSH is also the main endogenous nonprotein antioxidant because the active thiol group present in GSH is easily oxidized to oxidized glutathione (GSSG), which acts as a major physiologic free radical scavenger, participating in eliminating reactive oxygen species (ROS) and reactive nitrogen species (RNS) and facilitating electron transfer for enzymes such as glutathione peroxidase to reduce H_2_O_2_ to H_2_O [2]. Therefore, GSH plays an important role in maintaining the intracellular redox balance, participating in many detoxification reactions and preventing oxidative-stress-induced cellular damage [3]. An abnormal level of GSH is linked to a number of disease states such as cardiovascular diseases, immune diseases, aging diseases, and diabetes, and it is considered a marker for human disease [1]. A decrease in the level of GSH in the brain is observed in neuropsychiatric disorders such as autistic spectrum disorder, schizophrenia, bipolar disorder, and so on [4].

Owing to its importance, various analytical methods were developed to detect GSH. The tietze enzyme recycling assay was first reported in 1969 [5]. Other methods have emerged in recent years, including high-performance liquid chromatography (HPLC), capillary electrophoresis, mass spectrometry (MS), surface-enhanced Raman scattering (SERS), enzyme-linked immunosorbent assay (ELISA), and optical methods, including fluorescence and colorimetric approaches [6]. However, these methods have gradually revealed the disadvantages of complicated sample processing, time-consuming processes, and requirements for sophisticated equipment and trained technicians. With the increasing demand for miniaturization, especially for intracellular or in vivo analysis of GSH in complex biological systems, there is an urgent need to develop simple and efficient methods. Compared with the aforementioned methods, the electrochemical approach has the advantages of easy miniaturization, simple operation, fast analysis, and low cost, which meet the demand for in vivo real-time analysis and intracellular analysis.

Three strategies are presented for the electrochemical detection of biothiols: direct electro-oxidation, electro-reduction of disulfides, and indirect detection. It is not easy to ensure the electro-oxidation of GSH using traditional electrodes (for example, bare GCEs) due to their sluggish electron-transfer kinetics; therefore, mediators or electrocatalysts are required to decrease the overpotential. Until now, the promising electrocatalysts included carbon materials (such as CNT [7,8], mesoporous carbon [9], graphene oxide [10], graphene nanoribbon [11], and derived carbon [12,13]), Au nanomaterials [14,15], MN4 molecules [16,17,18,19,20,21], and TCNQ [22]. Another strategy for the electrochemical detection of thiols is based on the electro-reduction of the produced disulfides (that is, the conversion between a reduced and an oxidized state) [19,23]. The third method for thiol detection depends on the electrochemical response of the reaction product of thiol with quinine [24] or Cu^2+^ [25].

The main challenge with GSH detection is the chemical fouling caused by the oxidation product of GSSG and biofouling due to the non-specific absorption of biological macromolecules. Unlike CYS, which is abundant in serum, GSH is mainly present in cells. Therefore, it will be significant to achieve the intracellular or in vivo analysis of GSH with excellent antifouling properties for a better understanding of the relationship between GSH and disease, and it will also be helpful for the early diagnosis of diseases [26]. Direct electro-oxidation based on carbon materials is a promising method due to low cost and easy fabrication, modification, functionalization, and miniaturization. The carbon materials that have been reported for this purpose are limited to CNT, mesoporous carbon, and graphene oxide. Here, we demonstrated the unexpected oxidation of GSH on a glassy carbon electrode (GCE) by simple electrochemical treatment without any mediator or materials. The electrode can be reused to overcome the problem of chemical fouling, and it also showed excellent antibiofouling properties. This method will be a versatile approach used for carbon-based microelectrodes (for example carbon nanofiber) to achieve intracellular or in vivo analysis of GSH without any other modifier.

## 2. Experimental Section

### 2.1. Chemicals and Solutions

GSH, GSSG, uric acid (UA), ascorbic acid (AA), glucose, and bovine serum albumin (BSA) were purchased from Sigma-Aldrich (St. Louis, MO, USA). All other chemicals were of analytical reagent grade and used without further purification. PBS solutions at 0.1 M with different pH values were used as the background electrolytes and buffers for electrochemical detection.

### 2.2. Apparatus

X-ray photoelectron spectroscopy (XPS) was recorded on a Thermo ESCALAB 250 Xi spectrometer fitted with a monochromatic Al Kα X-ray source. Raman spectra were acquired with a high-resolution Raman spectrometer (LabRAM HR Evolution, HORIBA Scientific, Palaiseau, France) with a laser wavelength of 532 nm. The hydrophilicity test of different electrodes’ surfaces was evaluated using a static water-contact-angle measurement using a JC2000 Instrument (Shanghai Zhongchen Instrument Co., Ltd., Shanghai, China). Electrochemical experiments were performed on a CHI 750E electrochemical workstation with a conventional three-electrode system consisting of a working electrode, platinum coil auxiliary electrode, and a Ag/AgCl (saturated KCl) reference electrode. Electrochemical impedance spectra (EIS) were carried out in 0.1 M KCl containing 5 mM Fe(CN)_6_^3−/4−^ in the frequency range of 1 MHz to 0.1 Hz at 0.25 V.

### 2.3. Electrode Preparation and Modification

#### Preparation of Oxo-Functionalized Graphene

Oxo-functionalized graphene was formed in situ on the surface of the GCE using a two-step electrochemical treatment: First, the GCE was treated for a time period (0–800 s) in 0.1 M pH 7.0 PBS using an i-t curve at a high potential in the range of 1.6–1.8 V, which is denoted by EGO^1/potential^. Next, the prepared EGO^1/potential^ was reduced for 500 s in pH 4.0 acetic buffer using an i-t curve at a negative potential from −0.55 to −0.95 V and signed as ${\mathrm{EGO}}_{2/\mathrm{potential}}^{1/\mathrm{potential}}$. For example, ${\mathrm{EGO}}_{2/-0.85V}^{1/1.75V}$ refers to the GCE that was electrochemically oxidized at 1.75 V and subsequently reduced at −0.85 V. Unless otherwise specified, ${\mathrm{EGO}}_{2/-0.85V}^{1/1.75V}$ was used in the text.

## 3. Results and Discussion

### 3.1. Characterization of the Electrodes’ Interface

Graphite oxide, also called graphitic acid, has had a long history since it was discovered in 1840 [27]. A single layer of graphite oxide is called graphene oxide (GO), which possesses abundant oxygen-containing functional groups including hydroxyl, epoxy, and carboxyl groups, bonded to the graphene basal plane and edges with a C/O atomic ratio of around 2.0 [28]. Carboxylic acid is distributed at the edges, while phenol hydroxyl and epoxide groups are mainly present on the basal plane, making GO an amphiphile with strongly hydrophilic edges and a more hydrophobic basal plane [29]. Layers of oxidized graphite can be considered as a functionalized derivative of graphene with an intact carbon framework, which is termed oxo-functionalized graphene [27]. It was recently found that the residual oxygen-containing functional species present on graphene demonstrated high electrocatalytic activity towards the oxidation of glutathione [10], dihydroxybenzene isomers [30], L-methionine [30], uric acid, dopamine guanine, adenine [31], and ascorbic acid [32], as well as the reduction of oxygen [33,34] and polyphenol [35]. It is also significant for the in situ formation of oxo-functionalized graphene on the surface of electrodes. Although graphite is often used to synthesize GO, it is not suitable for the in situ electrochemical generation of oxo-functionalized graphene because severe expansion occurs soon after the electrochemical reaction starts [36]. GC is usually synthesized by the carbonization and graphitization of phenolic resin and exhibits a disordered graphitic structure in which nanosized graphite sheets are intertwined compactly with random orientations [36]. Unlike graphite, no obvious volume expansion was observed, which makes it possible for the in situ electrochemical generation of oxo-functionalized graphene on the surface of oxo-functionalized graphene. In addition, GCEs are the most widely used electrodes in electroanalysis and electrocatalysis.

The surface change in different electrodes could be differentiated by the naked eye (Figure 1). The color of the electrode changed from black to blue when the GCE was electrochemically oxidized. Upon subsequent electro-reduction, the electrode changed to a brown color. The hydrophilicity of different electrodes was characterized by a static water contact angle measurement (Figure 1). The water contact angles of the GCE, ${\;\mathrm{EGO}}_{}^{1/1.75V}\;$, and ${\mathrm{EGO}}_{2/-0.85V}^{1/1.75V}$ were found to be 88.9°, 46.29°, and 56.24°, respectively, indicating that the content of oxo-functionalized species were in the following: ${\mathrm{EGO}}_{}^{1/1.75V}$ > ${\mathrm{EGO}}_{2/-0.85V}^{1/1.75V}$ > GCE.

Raman spectroscopy is a powerful tool that characterizes the quality of the graphene framework and the defect. Crystalline graphene exhibits the G (~1585 cm^−1^) and the sharp 2D bands (~2700 cm^−1^) associated with the first- and second-order scattering of the E2g mode, respectively. The two-dimensional band stems from the stacking order of the crystalline graphite. However, the introduction of a high degree of disorder will reduce the 2D peak intensity and broaden the width, and the 2D peak will be split into four functions: 2D, 2D′, D + D′, and G*. These changes are also accompanied by the appearance of an intense peak centered at ~1350 cm^−1^, named the D band, which is related to defects in the graphite material such as bond-angle disorder, bond-length disorder, vacancies, edge defects, etc. [37]. In addition, in the low-wavenumber region, besides the G peak, three other shoulder peaks appeared, referred to as D″, D*, and D′. D″ is ascribed to amorphous phases because its intensity decreases when the crystallinity increases [38]. The D* band was reported with disordered graphitic lattices provided by sp2-sp3 bonds. Figure 2 shows the Raman spectra of the GCE (A),${\;\mathrm{EGO}}_{}^{1/1.75V}$ (B), and ${\mathrm{EGO}}_{2/-0.85V}^{1/1.75V}$ (C). The low-wavenumber region (the first-order band) could be fitted by five peaks, while the high-wavenumber region (the second-order band) was fitted by four peaks, attributed as shown in the figure labels. The fitted data about the different components are listed in Appendix A.

The content of D band (Appendix A) in ${\mathrm{EGO}}_{2/-0.85V}^{1/1.75V}$ was found to be higher than that of the GCE and ${\;\mathrm{EGO}}_{}^{1/1.75V}$ because graphite oxidation and the subsequent reduction seriously alter the basal plane structure of graphite [38]. The decrease in the I(D)/I(G) ratio, the right shift in G band in the full-width half maximum (FWHM), and the decreases in intensity of the 2D band were also found to be correlated with the oxidation level, which is consistent with the literature [39].

The chemical states of these electrodes were next investigated by XPS (Figure 3 and Figure 4, Appendix A). C 1s spectra (Figure 3) were fitted into four components: C−O at 285.8 eV, C=O at 287.0 eV, and O=C–O at 288.5 eV, as well as sp2 and sp3 hybridized carbon (C−C/C=C at around 284.5 eV) [40]. However, O-C=O was not observed in the GCE. Compared with the GCE, ${\mathrm{EGO}}_{}^{1/1.75V}$ shows a higher level of C=O and O-C=O species but lower C-O and C-C/C=C components due to the electrochemical oxidation. The subsequent electro-reduction results in a decrease in O-C=O and C=O contribution but an increase in the C-C/C=C and C-O contribution for ${\mathrm{EGO}}_{2/-0.85V}^{1/1.75V}$. Figure 4 shows the O 1s narrow spectra of different electrodes, decomposed into five peaks associated with O-C=O (~530.9 eV), C=O (~531.5 eV), C-OH (~532.6 eV), C-O-C (~533.5 eV), and chemisorbed oxygen and/or water (~535.0 eV), which might be linked with the hydrophilicity. For GCE, the contribution of O-C=O (only 1.8%) was almost not detected, and chemisorbed oxygen and/or water contribution was not observed. When GCE was electrochemically oxidized, the contribution of C=O and O-C=O species increased significantly, while the contribution of C-OH and C-O-C decreased. C-OH contribution increased again when the electrode was subsequently reduced. This is in agreement with the trends observed in the C1s spectra.

Fe(CN)_6_^3−/4−^ and Ru (NH_3_)_6_^2+/3+^ redox couples were also utilized to investigate the electrode interfaces using CV and EIS, as shown in Figure 5. The CVs of Fe(CN)_6_^3−/4−^ at different electrodes demonstrated a marked difference. The order of the redox currents was as follows: ${\mathrm{EGO}}_{2/-0.85V}^{1/1.75V}>\mathrm{GCE}>{\mathrm{EGO}}_{}^{1/1.75V}.$ The currents of ${\mathrm{EGO}}_{2/-0.85V}^{1/1.75V}$ are about 15 times that of ${\mathrm{EGO}}_{}^{1/1.75V}$. In addition, the charging (background) current of ${\mathrm{EGO}}_{}^{1/1.75V}$ is higher than that of the GCE. This is similar to the electrochemical properties of reduced graphene oxide. Unlike the case of Fe(CN)_6_^3−/4−^, the CVs of Ru (NH_3_)_6_^2+/3+^ exhibited a distinct trend. The GCE showed one pair of redox peaks between −0.1 and −0.2 V, which is found to be a diffusion-controlled process. In the case of ${\mathrm{EGO}}_{}^{1/1.75V}$ and ${\mathrm{EGO}}_{2/-0.85V}^{1/1.75V}$, besides the diffusion-controlled peaks, a pair of surface-controlled peaks was observed around −0.3 V. ${\mathrm{EGO}}_{}^{1/1.75V}$ possesses more abundant oxo-functionalized groups, for example, a negative carboxyl group, which repels Fe(CN)_6_^3−/4−^ but attracts Ru (NH_3_)_6_^2+/3+^. Fitted EIS curves showed that different electrodes’ charge transfer resistance R2 followed the order: ${\mathrm{EGO}}_{2/-0.85V}^{1/1.75V}>\mathrm{GCE}>{\mathrm{EGO}}_{}^{1/1.75V}\;$ (Appendix A).

It was considered that the anodic electrocatalytic oxygen evolution reaction of water occurred under a high positive potential, and reactive *OH, *O, and *OOH were formed. The absorbed reactive oxygen species reacted with the carbon lattice to form covalently bonded oxo-functionalized groups, and the rapid formation of a large amount of oxygen gas expanded the graphitic layers [28].

### 3.2. Electrochemical Sensing of GSH

The presence of some oxo-functionalized groups is beneficial for the electro-oxidation of GSH (Appendix A). An essential prerequisite for the efficient electro-oxidation of GSH is to optimize the oxo-functionalized species. Figure 6 shows the influence of oxidization potential on the CVs of 5 mM Fe(CN)_6_^3−/4−^ (A) and 5 mM GSH (B). We can see that the currents of redox probe Fe(CN)_6_^3−/4−^ decrease when the oxidation potential increases, suggesting the increased oxidation degree of the electrode. The current of Fe(CN)_6_^3−/4−^ reaches zero when the oxidation potential is about 1.8 V, indicating saturated oxo-functionalized groups. The effect of the oxidization potential on the electro-oxidation of GSH was investigated from 1.6 to 1.8 V (Figure 6 and Appendix A). The optimized oxidation potential was found to be 1.75 V. Other key factors were also investigated for the electro-oxidation including oxidation time (Appendix A), reduction potential (Appendix A), and pH of the solution (Figure 7 and Appendix A). The results showed that the optimized conditions for GSH electro-oxidation were found to be the oxidation potential of 1.75 V, the oxidation time of 500 s, the reduction potential of −0.85 V, and the solution pH of 4.5. GSH is a tripeptide molecule containing one NH_2_, two COOH, and one SH group with pKa1 = 2.12 (carboxylic acid of glutaminic acid), pKa2 = 3.59 (carboxylic acid of glycine), pKa3 = 8.75 (thiol of cysteine), and pKa4 = 9.65 (ammonium of glutaminic acid) [41]. It can be ionized into five species (H_4_A^+^, H_3_A, H_2_A^−^, HA^2−^, and A^3−^) by changing the solution pH. At pH 4.5, the dominant species is H_2_A^−^. Therefore, the anodic current under optimized conditions is attributed to the oxidation of H_2_A^−^. Gilbertson et al. investigated the interaction and catalytic oxidation mechanisms between GO and GSH and found that the synergism between the adjacent epoxide and hydroxyl groups on the surface contributed to the catalytic oxidation of GSH [42]. It was speculated that the pH value of the solution is linked with both the electroactivity of oxo-functionalities and the ionization of GSH. The following mechanism can be proposed for the electro-oxidation of GSH.
GSH = GS^−^ + H^+^(1)
GS → GS^•^ + e^−^(2)
2 GS^•^ → GSSG(3)

Next, the electrochemical detection of GSH was performed using the amperometric technique in a stirred 0.1 M PBS (pH 4.5) solution on ${\mathrm{EGO}}_{2/-0.85V}^{1/1.75V}$ at an applied potential of 0.2 V (Figure 8). The response current increased with the addition of GSH, and a linear response was observed over a concentration range from 1 μM to 1.09 mM with a calculated LOD of 0.3 μM. When the concentration was higher than 1.09 mM, the response was unstable and deviated from linearity. This is caused by chemical fouling of the electrode surface derived from the oxidation product GSSG. The current responses of the electrode to various concentrations of GSH in a static 0.1 M PBS (pH = 4.5) were also performed (Appendix A). The current increased linearly with the concentrations of GSH in a wide range from 1 μM to 2 mM.

### 3.3. Chemical Fouling Recovery and Antibiofouling Properties

It was reported that the passivation of the electrode often occurred for the electrochemical oxidation of thiols [10], which might be caused by the binding of the sulfur moiety to the surface [43]. Chemical fouling recovery of the electrode was also investigated for the oxidation of GSH (Figure 9). When the ${\mathrm{EGO}}_{2/-0.85V}^{1/1.75V}$ was directly scanned in the presence of GSH without scanning in the background solution before, the first cycle showed a distinct difference compared with the sequential cycling. Two oxidation processes waves at 0.15 V and 0.2 V were observed in the first cycle, which may be attributed to the oxygen-containing functional groups (quinone-like functional groups) and the edge plane-like defective sites [44]. The oxidation peaks in the sequential cycles decreased gradually due to chemical fouling. Upon the second and third treatments of the electrode by electrochemical oxidation and reduction, the response was completely recovered, indicating that the electrode could be reused by electrochemical renewal. In fact, a protective layer is usually needed to ensure the integrity of the active layer when a microelectrode is used to penetrate into tissues in case of in vivo analysis. We proposed that this active layer could be formed in situ and regenerated in tissue by the electrochemical treatment at physiological pH. As a result, the protective layer would not be used with a process simplification.

For the analysis of biological samples, conventional electrodes often undergo biofouling associated with non-specific absorption of biological macromolecules (especially proteins such as BSA), which hinders electron transfer and thus leads to a rapid loss of sensitivity. In order to evaluate the practical application of the electrode in biological fluids, an antibiofouling test was also carried out. The Fe(CN)_6_^3−/4−^ redox couple was utilized to investigate the binding of electrodes with BSA and the antibiofouling property (Figure 10). ${\mathrm{EGO}}_{2/-0.85V}^{1/1.75V}$ has a larger electro-active surface area than that of the unmodified GCE due to its wrinkled structure. However, the carboxylic group presented on ${\mathrm{EGO}}_{2/-0.85V}^{1/1.75V}$ is negative and repels the negative Fe(CN)_6_^3−/4−^. As a result, the current of Fe(CN)_6_^3−/4−^ at ${\mathrm{EGO}}_{2/-0.85V}^{1/1.75V}$ is approximately identical to that at the unmodified GCE. The electrodes before and after immersing in BSA exhibited distinct differences in the Fe(CN)_6_^3−/4−^ solution. The GCE showed an obvious decrease in redox currents after immersing in the 10 mg/mL BSA solution for 30 min due to the biofouling of BSA. However, an increase in redox currents was observed upon immersing in the BSA solution for ${\;\mathrm{EGO}}_{}^{1/1.75V}$ and ${\mathrm{EGO}}_{2/-0.85V}^{1/1.75V}$, which might be due to the presence of oxygen-containing functional groups on the electrode surface. It was unexpected to observe that the current of graphene oxide electrode immersed in the BSA solution was about 25 times higher than that of pure graphene oxide electrode [31]. This indicated that the BSA coating was formed on the electrode surface and the presence of BSA did not hinder but increased electron transfer and mass transport. We think that many ordered channels might be formed on the electrode surface due to the multiple interactions between BSA and the oxo-functionalized graphene including hydrophilic interaction, electrostatic interaction, and hydrogen-bonding interaction. Both the oxo-functionalities and basal graphene played an important role in the binding of the ordered BSA. It was considered that the ordered channel improved the mass transport of small molecules. However, in the case of the bare GCE, disorganized absorption of BSA occurred on the surface, which hindered electron transfer and mass transport. The results showed that the presence of BSA did not influence electron transfer at ${\mathrm{EGO}}_{2/-0.85V}^{1/1.75V}$, indicating good anti-biofouling properties.

## 4. Conclusions

This study demonstrates a novel electrochemical method for the sensitive determination of GSH based on a simple and reusable GCE treated by the electrochemical oxidation and reduction method with chemical fouling recovery and antibiofouling properties. Various characterization methods showed that oxo-functionalized graphene was formed in situ on the surface of the GCE. This approach will be a promising tool to treat carbon-based microelectrodes for potential intracellular GSH monitoring or in vivo analysis in the brain, which will be helpful to better understand the complex biochemistry of this molecule in health and disease because GSH depletion in the brain is found in patients with neurodegenerative diseases and the GSH level varies by brain region [45,46].

## Figures and Tables

**Figure 1 antioxidants-12-00008-f001:**
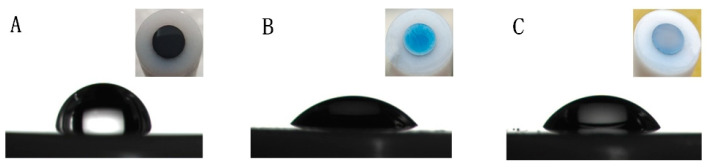
Images of contact angles of water drops deposited on GCE (**A**),${\;\mathrm{EGO}}_{}^{1/1.75V}$ (**B**), and ${\mathrm{EGO}}_{2/-0.85V}^{1/1.75V}$ (**C**).

**Figure 2 antioxidants-12-00008-f002:**
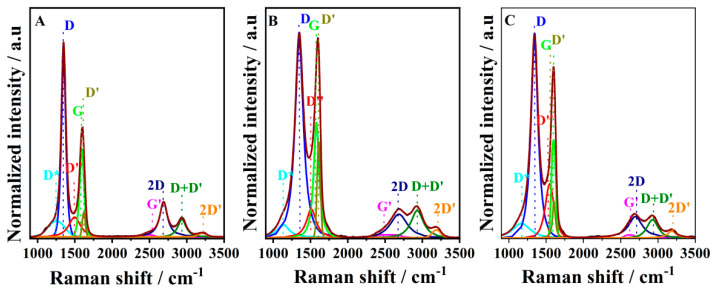
Raman spectra of GCE (**A**),${\;\mathrm{EGO}}_{}^{1/1.75V}\;$ (**B**), and ${\mathrm{EGO}}_{2/-0.85V}^{1/1.75V}$ (**C**).

**Figure 3 antioxidants-12-00008-f003:**
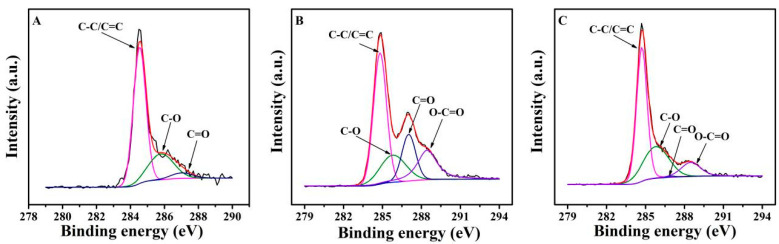
C1s spectra of GCE (**A**),${\;\mathrm{EGO}}_{}^{1/1.75V}$ (**B**), and ${\mathrm{EGO}}_{2/-0.85V}^{1/1.75V}$ (**C**).

**Figure 4 antioxidants-12-00008-f004:**
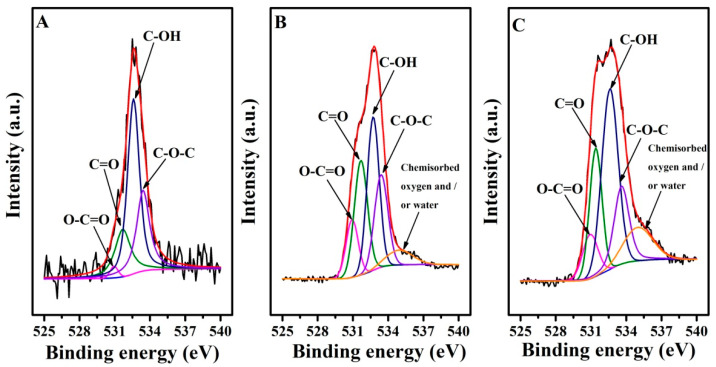
O1s spectra of GCE (**A**),${\;\mathrm{EGO}}_{}^{1/1.75V}$ (**B**), and ${\mathrm{EGO}}_{2/-0.85V}^{1/1.75V}$ (**C**).

**Figure 5 antioxidants-12-00008-f005:**
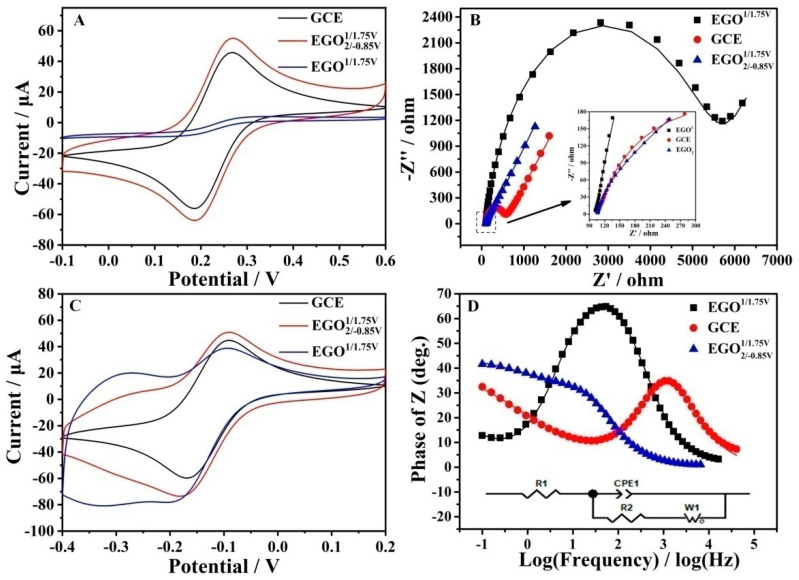
CVs (**A**,**C**), Nyquist diagrams (**B**) and phase angle diagrams vs. the log of frequency of Bode plots (**D**) for GCE,${\;\mathrm{EGO}}_{}^{1/1.75V}$, and ${\mathrm{EGO}}_{2/-0.85V}^{1/1.75V}$ in 5 mM Fe(CN)_6_^3−/4−^ (**A**,**B**,**D**) and Ru (NH_3_)_6_^2+/3+^ (**C**) redox couples at a scan rate of 25 mV/s.

**Figure 6 antioxidants-12-00008-f006:**
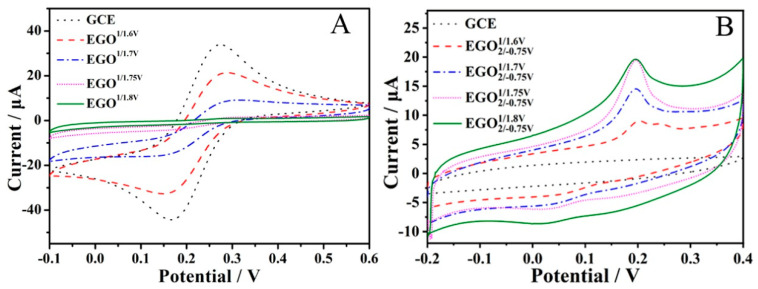
CVs of different EGO^1/potential^ and ${\mathrm{EGO}}_{2/-0.75}^{1/\mathrm{potential}}$ in 5 mM Fe(CN)_6_^3−/4−^ (**A**) and 5 mM GSH (**B**) in pH 5.0 PBS.

**Figure 7 antioxidants-12-00008-f007:**
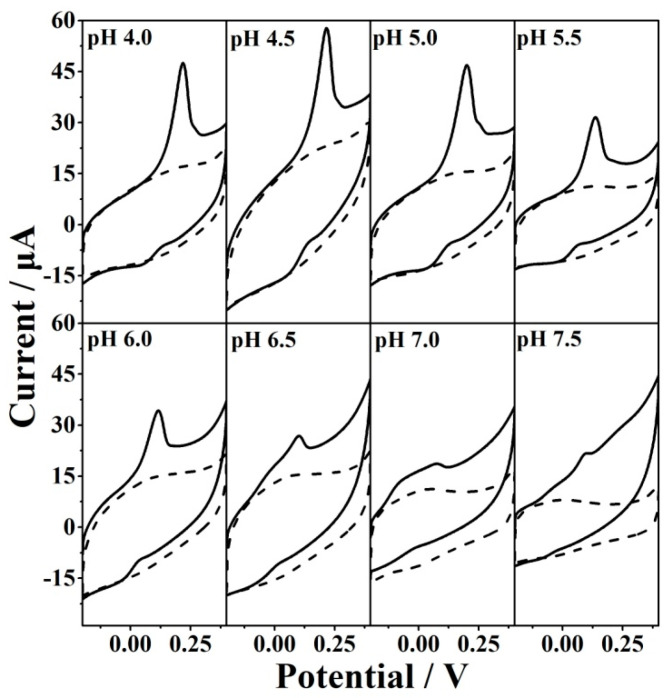
CVs of ${\mathrm{EGO}}_{2/-0.85V}^{1/1.75V}$ in the presence and absence of 5 mM GSH in PBS with different pHs.

**Figure 8 antioxidants-12-00008-f008:**
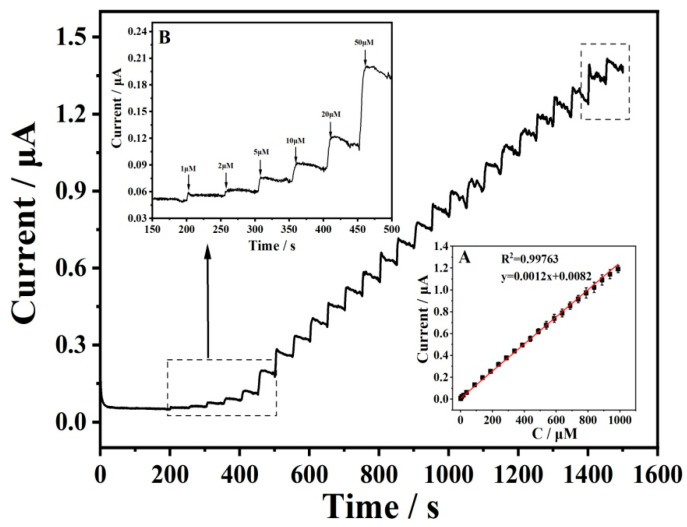
Amperometric response of ${\mathrm{EGO}}_{2/-0.85V}^{1/1.75V}\;$ to the successive injection of GSH in a stirred 0.1 M PBS (pH 4.5) at an applied potential of 0.2 V. Inset (**A**) is the calibration curve for the steady-state current upon the addition of different GSG levels. Inset (**B**) is the amperometric response at low concentration.

**Figure 9 antioxidants-12-00008-f009:**
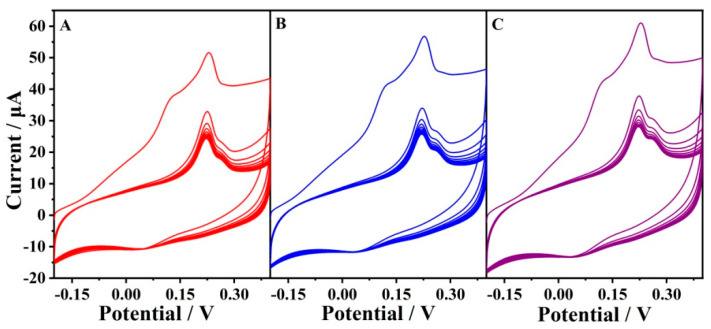
Reusable ${\mathrm{EGO}}_{2/-0.85V}^{1/1.75V}$ for the CV test of 5 mM GSH by a repeatable electrochemical oxidation and reduction treatment. (**A**) The first treatment; (**B**) the second treatment; (**C**) the third treatment.

**Figure 10 antioxidants-12-00008-f010:**
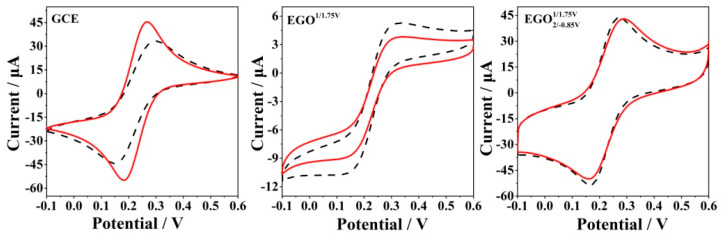
CVs of GCE, ${\;\mathrm{EGO}}_{}^{1/1.75V}$, and ${\mathrm{EGO}}_{2/-0.85V}^{1/1.75V}$ in the presence (dashed line) and absence (solid line) of 10 mg/mL BSA in 5 mM Fe(CN)_6_^3−/4−^.

## Data Availability

The authors confirm that the data supporting the findings of this study are available within the article [and/or] its Appendix A.

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
