# Peer review of "In Situ Electrochemical Formation of Oxo-Functionalized Graphene on Glassy Carbon Electrode with Chemical Fouling Recovery and Antibiofouling Properties for Electrochemical Sensing of Reduced Glutathione"

_antioxidants, 2022, doi:10.3390/antiox12010008_

Round 1

Reviewer 1 Report

In this manuscript, the authors proposed a novel electrochemical method useful for  in vivo analysis of intracellular reduced glutathione (GSH), considered a marker for human diseases, helpful for the early diagnosis of various diseases.

The authors developed a simple GSH sensor, with good anti-biofouling properties, respectively a  reusable carbon-based electrode (GCE)  treated by electrochemical oxidation and reduction method. Different methods were used in order to characterize their microelectrodes, respectively to confirm that oxo-functionalized graphene was in situ formed on the surface of GCE and to evaluate the electrochemical detection of GSH (including the application of the electrode in biological fluids.

However, I consider the following additions important:
3.1. Characterization of electrodes (electrode interface)

I suggest completing the characterization of electrode interface by X-ray diffraction (XRD) in order to highlight the presence of oxo-functionalized graphenes. Also through  transmission electron microscopy / scanning electron microscopy (SEM / TEM).

3.2. Electrochemical sensing of GSH
- Voltametric method: to be completed with the calibration curve (for CV) and with the determination of the detection limit  (cyclic voltammetry is presented, highlighting only the variation with pH!) 

- Amperometric method: to standardize the notations, respectively to check the correspondence between the text/figure data (since the text mentions the performance of amperometric measurements at 0.25V  potential, and in figure 8 it appears 0.20V).

 I consider this article deserves publication in the Journal, and I recommend the publication of this manuscript taking into consideration the mentioned sugestions.

Reviewer 2 Report

The subject addressed by the authors may be of interest to Antioxidants, but the article is poorly written and in many places difficult for readers to understand.

An important problem is that the practical usefulness of the developed sensor is not clear from the presented results. My opinion is that to state that the sensor can be applied as a microelectrode for in vivo tests, there must be a microelectrode. But in the present study it is not about the microsensor. In addition, tests on real samples, which imitate the conditions in the body, have not been performed, so it is not known how the scitivated electrode would behave in real, complex matrices. Such tests would increase the value of the study.

The introduction is quite chaotically presented and needs to be restructured and shortened so that it is correlated with the experimental data and with their inclusion in the state of the art on this topic

In section 3.3. Chemical fouling recovery and antibiofouling properties, the data presented are not sufficiently explained. The authors are asked to explain how the fouling recovery procedure is performed and how this step can be performed for sensors intended for in vivo tests?

In the section where the antibiofouling results are presented, it is specified that "GCE showed obvious decrease in redox currents after immersing in 10 mg/mL BSA solution due to biofouling of BSA. However, increase in redox currents were observed upon immersing in BSA solution for EGO1 and EGO1/2”. From the CV curves presented in Figure 10, it is not clear that the signal on EGO1 and EGO1/2 increases after contact with the BSA solution, thus, more explanations should be provided by authors.

The authors are asked to explain in more detail the practical applicability of the study.

The manuscript must be carefully revised by the authors. There are mistakes in writing (missing letters, extra letters, duplicated words), but also in the English language. In addition, there are many unfinished sentences that must be checked and revised. The opinion and help of a native English speaker would be beneficial.

In this form, the manuscript cannot be recommended for publication. Major revision is necessary, for which the authors are asked to consider the observations made

Round 2

Reviewer 2 Report

Thanks to the authors for the effort to improve the quality of the manuscript and answer. After the review, the quality of the manuscript has improved, at least as far as the English language is concerned.
However, the answer to some questions did not convince me.
Thus, for the first question, the authors justified with a figure the applicability of the method for the development of implantable sensors, but in the voltammogram there is a reduction signal for GSH. In the manuscript, the electrochemical transformation of GSH is presented by the oxidation signal (see Figure 7). How do the authors justify these differences? Did they use another potentiostat?
Also, the answer to question 4 is not complete. From Figure 10 it can be seen that the unmodified GCE signal is approximately identical to that of the functionalized electrode (first voltammogram on the left, compared to the last voltammogram on the right). In this case, what is the advantage of the author presentation method?
In conclusion, in this form I cannot recommend the publication of the manuscript.
